# Numerical Simulation on Backfilling of Buried Pipes Using Controlled Low Strength Materials

Hao Liu, Yiqi Xiao, Kaixin Liu 🄳, Youzeng Zhu and Peng Zhang *

Faculty of Engineering, China University of Geosciences, Wuhan 430074, China; cugwuhanly@163.com (H.L.); hhlxyq1172195103@163.com (Y.X.); 15207162405@163.com (K.L.); youzengzhu@163.com (Y.Z.)
* Correspondence: cugpengzhang@163.com

**Abstract:** The backfill quality of a pipeline has an important influence on pipeline operation. When loose backfill is used, the pipeline may be damaged after short term operation. In this study, the numerical simulation analysis of buried pipes was carried out under three conditions: loose backfill around the pipe, dense backfill, and controlled low strength materials (CLSM) backfill. The effects of narrow trench backfilling using CLSM on the force and deformation of pipelines were studied. The results showed that When CLSM was used for buried pipe backfilling, the pressure on the top of the pipe and on the side of the pipe was significantly reduced. When the surface pressure was 200 kPa, the radial displacement at the top of the pipe was only 0.6 mm. Compared with the dense backfill of the pipe, the radial displacement of the pipe top was reduced by 82.9%, which greatly reduced the deformation of the pipe. CLSM backfilling is a good way to protect the pipeline. The pressure is uniformly applied around the pipe, and the circumferential strain around the pipe is greatly reduced. Pipelines backfilling with CLSM for buried flexible pipes has good mechanical properties and it is expected to be applied to engineering practice.

**Keywords:** buried pipes; controlled low strength materials; numerical simulation; static load; flexible pipes; mechanical properties

## 1. Introduction

The backfill compaction quality of pipelines has an important influence on the safe operation of pipelines [1,2]. At present, existing water supply and drainage pipelines are mostly backfilled with solid particles. When groove backfill is conducted, the bottom area of the pipeline is narrow, and there is a backfill dead angle in the effective support foot area of the pipeline. Due to the difficulty of using compaction equipment, the compaction quality is difficult to guarantee. Pipelines may be damaged after long term operation. Once the pipeline leaks, the surrounding soil will gradually lose, which leads to frequent road cracking, settlement and collapse accidents.

Controlled low strength materials (CLSM) is a self-compacting and high fluidity low strength material defined by ACI 229 [3]. It can effectively backfill the area that cannot be compacted by large compaction equipment. Different from the conventional backfill method, it only needs to be filled with fluidization, which greatly saves resources and costs. At present, CLSM has been extensively studied in China and abroad. In general, CLSM is mainly composed of cement, fly ash, sand and cement. In recent years, many scholars have studied the feasibility of using green and sustainable materials as aggregates to prepare CLSM. Young-sang Kim et al. [4] and Bhaskar Chittoori et al. [5] successfully prepared CLSM using waste soil instead of aggregates. Etxeberria Miren et al. [6] used construction waste to prepare CLSM using only 110 kg of water per cubic meter. The same acceptable CLSM performance was obtained as in the control group. Rui Xiao et al. [7] successfully prepared CLSM based on the volcanic ash reaction of waste glass powder and hydrated lime, which provided a way for waste glass to be used in the construction

industry. Yeong-Nain Sheen et al. [8] studied slag preparation CLSM and found that an increase in slag content can improve processability and reduce strength. Lulu Liu, et al. [9] used recycled tire rubber-sand mixtures as lightweight backfill and found that the effective use of waste tires can reduce environmental pollution. Naganathan et al. [10] studied the comprehensive use of industrial waste incineration bottom ash to prepare CLSM and found that excessive use of bottom ash will have the problem of water secretion.

In addition, some scholars have studied the application of CLSM to construction engineering. Pier Paolo et al. [11] applied CLSM to tunnel pavement for laboratory and field test studies. It can be paved as a pavement but must meet certain acceptance criteria. Tan Manh Do et al. [12] used marine dredged soil to make CLSM for geothermal systems, which could reduce the total cost by up to 37%. Kyung-Joong Lee et al. [13] conducted small-scale indoor experiments on controlled low strength material from regenerated soils and concluded that CLSM may be a potential option for stabilizing underground piping systems. A. Blanco et al. [14] proposed a method to optimize CLSM design that could be applied to the backfilling of narrow trenches. A. Barghi Khezerloo et al. [15] use CLSM mixtures as pipe backfills. The final backfill took less than a day and the piping performance was satisfactory.

At present, there are many studies on the preparation of CLSM using various types of aggregates, but there are few studies on the application of CLSM to buried pipeline backfilling. The force deformation characteristics of the pipe are still unclear. For buried pipes, it is quite necessary to study the arching effect between them and the surrounding soil. Figure 1 shows a schematic diagram of the arching effect of rigid and flexible pipes in soil. Buried pipes share surface loads with surrounding soils. For rigid pipes, negative earth arches are formed above the pipe, while for flexible pipes, positive earth arches are formed above the pipe [16,17], reducing the force above the pipe. For the backfilling of buried pipelines using CLSM, the "pipe soil action" between it and the pipeline is not yet known. Therefore, this study conducted a numerical simulation study on the backfilling of buried pipelines using CLSM. The force and deformation characteristics of buried pipes were studied. The effects of soil compaction quality and load size around the pipe on the earth pressure and annular strain of the pipe were comprehensively analyzed. It will provide a good motivation for the application of CLSM to engineering practice.

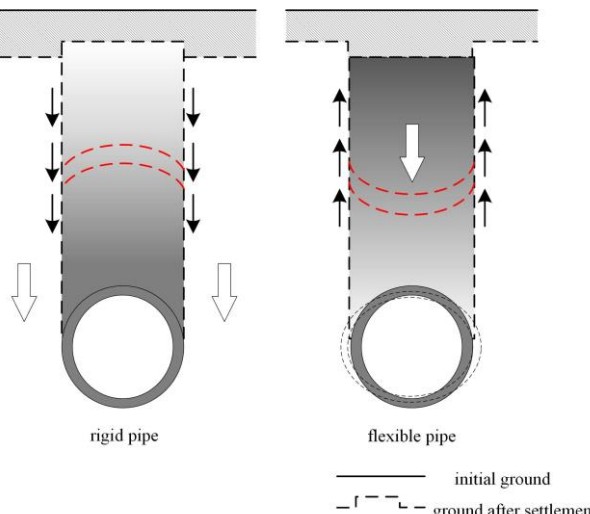

**Figure 1.** Soil arching effect of rigid and flexible pipes in soil.

## 2. Pipe and Backfill Material Parameters

### 2.1. Testing of Pipe Ring Stiffness

In order to accurately understand the physical properties of the pipeline and provide corresponding parameters for numerical simulation, the ring stiffness of the pipeline was tested. The ring stiffness of PE pipe was measured according to the "GB/T9647-2003

Determination of Ring Stiffness of Thermoplastic Pipes" [18], and the true elastic modulus of the pipe was reversed according to the formula. The test instrument included PE pipes with a length of 300 mm, an electronic ring stiffness machine, etc.

The operator adjusted the pipe position before pressing the pipe to ensure that the long shaft of the PE pipe is parallel to the machine and placed in the center of the ring stiffness machine. A deformation tester was set up in each specimen to measure the pipe displacement. The specimen was compressed at a speed of 10 mm/min according to the specification until the pipe deformation reaches at least 0.03 d, recording the force value and deformation amount. Schematic diagram of PE pipe ring stiffness test is shown in Figure 2. Equation (1) is used to calculate the ring stiffness of the pipe:

$$S = \frac{(0.0186 + 0.025Y/d)F}{LY} \tag{1}$$

where:

$F$—Pressure value (kN) when the pipe is deformed by 3.0%;
$L$—specimen length in meters (m);
$Y$—pipe deformation (m);
$S$—ring stiffness of the specimen in kN/m$^2$.

Three identical tubes were taken for testing, and the average of the three was used as the final value of the ring stiffness. The specific test results and summary are shown in Table 1.

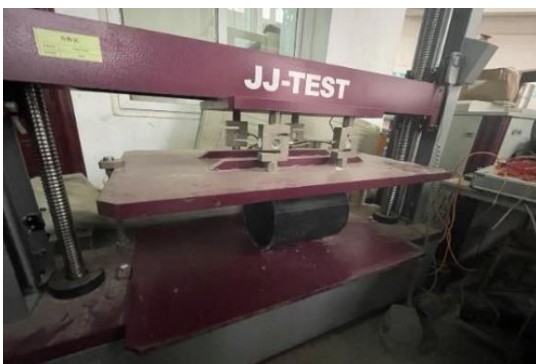

**Figure 2.** Schematic diagram of PE pipe ring stiffness test.

**Table 1.** Ring stiffness of PE pipe.

| Test Number | 1 | 2 | 3 |
|---|---|---|---|
| Compressive force value/kN | 1.13 | 1.10 | 1.15 |
| Pipe ring stiffness/(kN/m$^2$) | 7.41 | 7.22 | 7.54 |
| Mean ring stiffness/(kN/m$^2$) | | 7.40 | |

According to the "Manual of Structural Design of Water Supply and Drainage Engineering" [19], the relationship between the ring stiffness of PE pipes and the modulus of pipes is shown in Equation (2):

$$S_p = \frac{E_p}{12} \left( \frac{t_0}{D_0} \right)^3 \tag{2}$$

The formula for calculating the elastic modulus of the pipeline is as follows (3):

$$E_p = 12 S_p \left( \frac{D_0}{t_0} \right)^3 \tag{3}$$

where:

$E_p$—elastic modulus/MPa of pipe;
$S_p$—Tube ring stiffness/MPa;
$D_0$—Pipe diameter/mm;
$t_0$—Calculation of wall thickness/mm of pipe

### 2.2. Performance Test of Controlled Low Strength Material

The CLSM mixture ratio was tested in Table 2. Engineering excavation soil and sand were used as fine aggregate, and the engineering excavation soil was replaced by 10%, 20% and 30% of sand, cement and fly ash as cementitious materials.

**Table 2.** Mixture ratio of test.

| Excavation Soil Content | Material (kg/m$^3$) | | | | |
|---|---|---|---|---|---|
| | Cement | Fly Ash | Water | Sand | Excavation Soil |
| 10% | 75 | 200 | 340 | 1192.5 | 132.5 |
| 20% | 75 | 200 | 370 | 1060 | 265 |
| 30% | 75 | 200 | 425 | 927.5 | 397.5 |

Natural river sand as fine aggregate is medium sand. The fineness modulus is 2.4. Excavation soil comes from the Wuhan project. In this study, a 5 mm sieve was used for screening, and large particles were screened out. The liquid limit of excavated soil is $\omega l = 15.3$, the plasticity index is Ip = 17.3, and the natural moisture content is 20.18%. In this study, Portland cement with compressive strength of 42.5 MPa and Class F primary fly ash were used.

The unconfined compressive strength test used cube specimens with dimension of 70.7 mm × 70.7 mm × 70.7 mm. After 28 days' curing of CLSM, the strength test was carried out in the YAW-4605 pressure testing machine. CLSM's stress–strain test used size of rectangular specimen, which was 70.7 mm × 70.7 mm × 210 mm. After 28 days' curing in the curing box, the stress–strain test was carried out using a servo pressure testing machine. Unconfined compressive strength and stress–strain tests are implemented in accordance with "GB/T 50081−2019. Standard Test Method for Physical and Mechanical Properties of Concrete" [20].

The stress–strain curve of CLSM is shown in Figure 3. When the amount of excavation soil is 10%, CLSM reaches the peak stress, then the specimen is destroyed, and the stress decreases rapidly. With the increase of excavation soil content in CLSM, the plastic strain of CLSM gradually increases. CLSM reaches peak stress drop more slowly, has a large plastic deformation, and will not be destroyed immediately. CLSM reaches peak stress drop more slowly. The material has a large plastic deformation and will not be destroyed immediately. It can be concluded that increasing the clay content in the aggregate can increase the plastic deformation of CLSM.

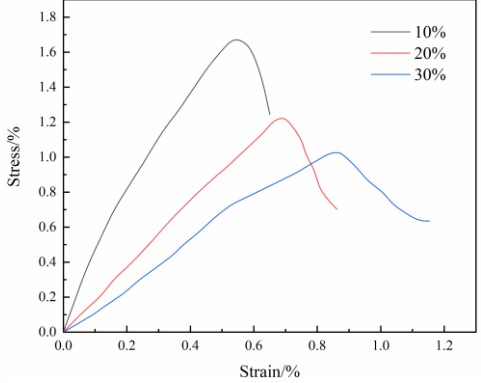

**Figure 3.** The stress–strain curve of CLSM.

For the comparison of CLSM's unconfined compressive strength and elastic modulus, as shown in Figure 4, the modulus of elasticity and compressive strength change trend is the same. When the excavation soil content is 10%, CLSM has the largest modulus of elasticity at 290 MPa. When the excavation soil content in CLSM increased to 30%, the elastic modulus of CLSM is only 130 MPa. In the numerical simulation that follows, CLSM with an excavation soil amount of 20% is used, and its modulus of elasticity is 202 MPa.

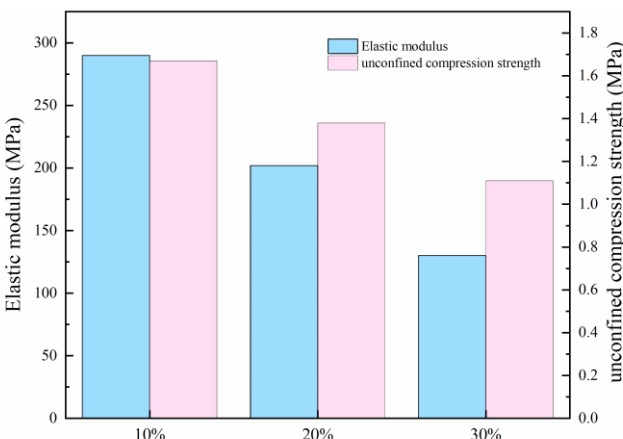

**Figure 4.** Comparison of the elastic modulus of CLSM with the unconstrained compressive strength.

## 3. Numerical Simulation Model

In this study, the influence of backfill quality, backfill materials and the surface pressure on the mechanical properties of pipelines under static load was studied by numerical simulation. This study used Abaqus finite element software to perform a numerical simulation. The calculation used a calculation model for planar strain. Its boundary condition was to limit the horizontal displacement of the two sides, and the bottom was a fixed boundary. The model test size was set to 1.5 × 1.2 m (width × height), as shown in Figure 5.

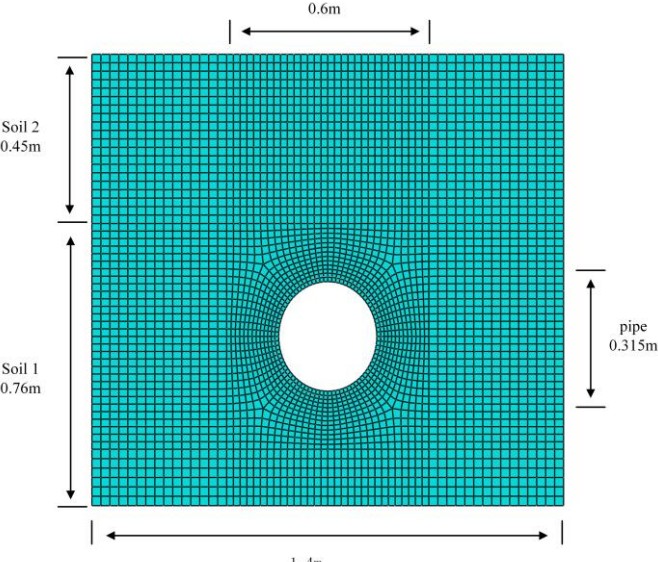

**Figure 5.** Schematic diagram of the numerical simulation.

In the numerical model calculation, the process of applying loads above the pipe was simulated by applying pressure step by step to the surface, and the applied load range was within 0.6 m of the surface directly above the pipe. The loads were loaded step by step, and the load were increased from 100 kPa to 150 kPa to 200 kPa.

PE pipe with a diameter of 315 mm and a wall thickness of 12 mm was used in the test. In the finite element calculation, the pipe is regarded as an elastic body, and the elastic modulus E of the pipe is determined to be 820 MPa. The Poisson ratio is 0.4 according to the ring stiffness test. The pipe–soil contact adopts surface–surface contact, with the pipe surface as the main control surface, the soil surface as the subordinate surface. The friction coefficient between the pipe and the soil is 0.4.

Soil 1 used in the numerical simulation uses three conditions. They are compact backfill, loose backfill, and CLSM backfill. The backfill height is 0.76 m. The upper part is backfilled with a sandstone material (Soil 2) with a backfill height of 0.45 m.

The material properties of sand refer to the material parameters in the Xiao Chengzhi [21] study. The Mohr–Coulomb constitutive model is used for finite element numerical calculations for sand and CLSM. The cell meshing uses quadrilateral elements. The specific material parameters are shown in Table 3.

**Table 3.** Material properties of numerical simulation.

|  | Condition | Density/(kg/m³) | Elastic Modulus/MPa | Poisson Ratio/μ | Dilation Angle/° | Cohesion/kPa |
|---|---|---|---|---|---|---|
|  | dense backfill | 1650 | 30 | 0.3 | 37.8 | 12 |
| Soil 1 | loose backfill | 2000 | 5 | 0.3 | 20 | 10 |
|  | CLSM backfill | 1800 | 202 | 0.3 | 35 | 300 |
|  | Soil 2 | 1650 | 10 | 0.3 | 20 | 15 |

## 4. Results

### 4.1. Verification of Modeling Techniques

In order to verify the correctness of the modeling techniques, the laboratory model test in Xiao Chengzhi's study [21] is numerically simulated. Based on the plane strain test, the laboratory test was carried out in a model box with length, width and height of 120 cm, 40 cm and 98 cm, respectively. The load above the pipeline is loaded by a rigid plate with thickness of 2 cm, width of 12 cm and length of 38 cm. In the model experiment, the outer diameter D of the pipeline is 110 mm, the thickness t of the pipeline is 53 mm, and the density ρ is 0.97 g/cm³. Based on the modeling techniques in this paper, the correctness of the modeling techniques were verified by numerical simulation. The soil pressure on the pipe side in the model experiment was compared with the simulation results. Extracting the earth pressure S1, S2 and S3 of the pipe side, as shown in Figure 6. When the surface pressure is 120 kPa and 240 kPa, the results are shown in Figure 7. The numerical simulation results are in good agreement with the model test results, which verifies the correctness of the numerical modeling techniques in this study.

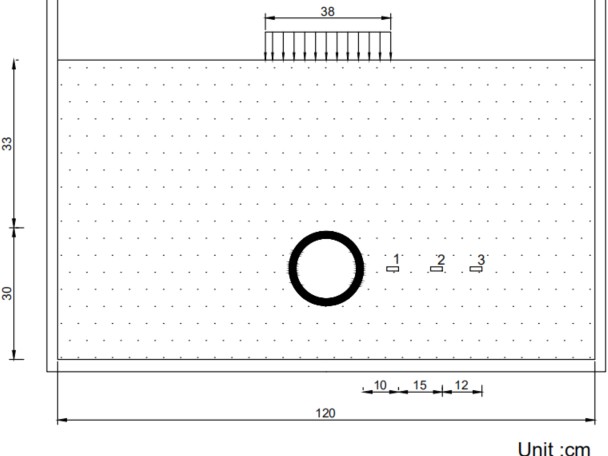

**Figure 6.** Diagram of earth pressure extraction.

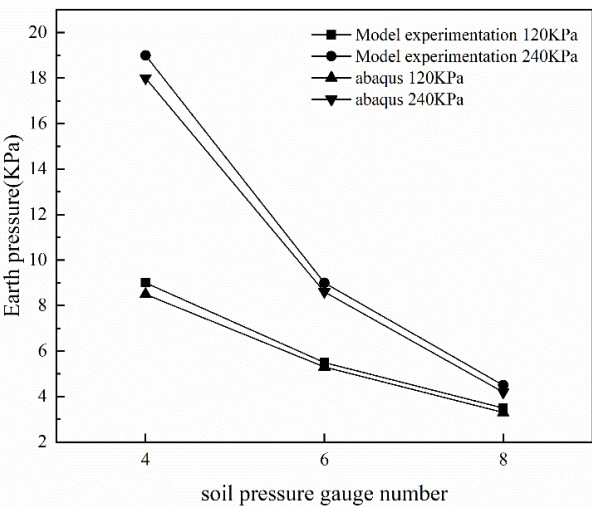

**Figure 7.** Lateral earth pressure of the pipe.

### 4.2. Deformation Nephogram of Buried Pipes

When the surface pressure is 200 kPa, the deformation and displacement of buried pipe and soil are shown in Figures 8 and 9. When dense backfill is adopted, the surface load is jointly borne by the pipe and the soil, forming the state of pipe–soil interaction [22,23]. However, using CLSM backfill, the force transferred from the upper surface load is mostly borne by CLSM, and the pipeline is almost unforced. CLSM protects the pipeline well.

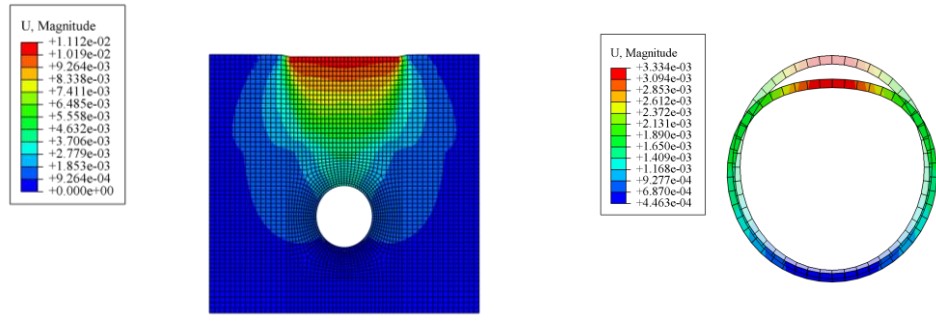

(**a**) displacement of surface soil        (**b**) deformation of the pipe

**Figure 8.** Deformation nephogram of dense backfill.

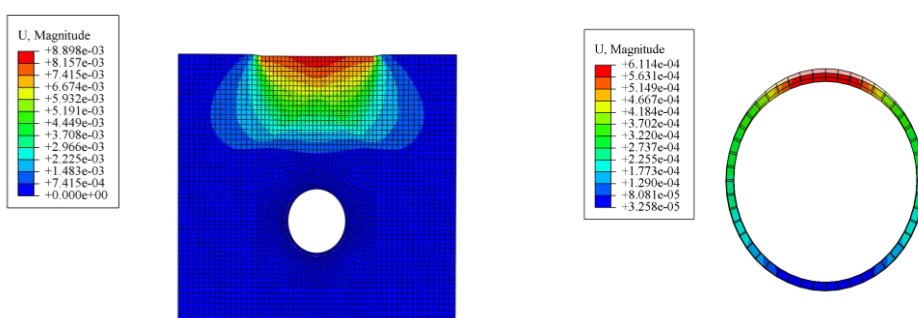

(**a**) displacement of surface soil        (**b**) deformation of the pipe

**Figure 9.** Deformation nephogram of CLSM backfill.

### 4.3. Analysis of Soil Pressure Characteristics

For the buried pipeline, numerical simulation is carried out under three conditions of loose backfill, dense backfill and CLSM backfill. The stress data of soil peri-pipe under different working conditions were extracted, and the change of soil pressure around pipe

was studied. The vertical earth pressure of the pipe is S1 and S2, and the interval between the two is 15 cm. The horizontal earth pressure on the right side of the pipe is S3 and S5, and the vertical earth pressure is S4 and S6. The interval between the two is 15 cm. The arrangement of earth pressure monitoring around the pipe is shown in Figure 10.

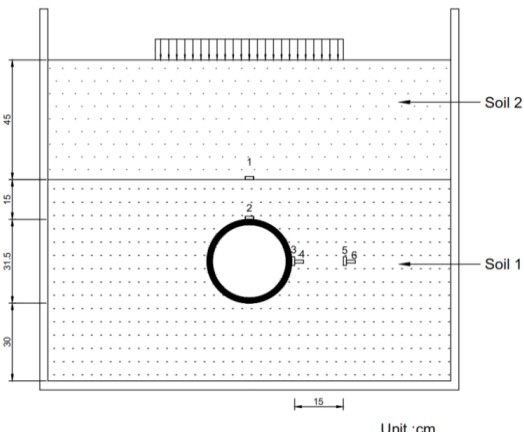

**Figure 10.** Schematic diagram of the location of earth pressure monitoring.

Figure 11 shows the earth pressure at various locations in the soil at a surface pressure of 200 kPa. Comparing the loose backfill around the buried pipe with the dense backfill, the earth pressure change trend is basically similar at each location. Under the condition of loose backfill, the vertical earth pressure S2 at the top of the pipe and the horizontal earth pressure S3 on the pipe side increased compared with the dense backfill. Vertical earth pressure increased by 8% and horizontal earth pressure S3 increased by 30.6%. The vertical earth pressure of the pipe top and the horizontal earth pressure on the pipe side of the pipe are not conducive to the force of the pipeline. Compared with the use of CLSM for buried pipe backfilling, the vertical earth pressure S1 at 15 cm above the pipeline increased, compared with the dense backfill around the pipeline increased by 6%. The vertical earth pressure S2 and the horizontal earth pressure S3 of the pipe side are significantly reduced. When using CLSM for buried pipe backfilling, the vertical earth pressure S1 at 15 cm above the pipe increases. Compared to the dense backfill around the pipe, it increased by 6%. The vertical earth pressure S2 and the horizontal earth pressure S3 of the pipe side were significantly reduced.

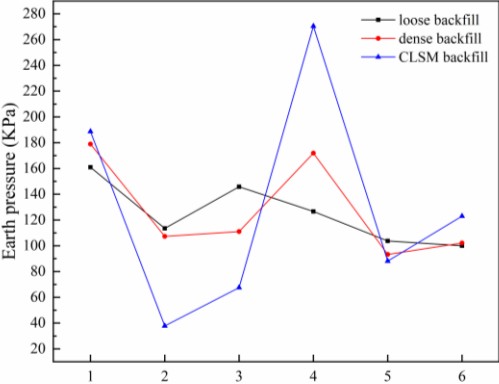

**Figure 11.** The earth pressure at each position when the static load pressure is 200 kPa.

Figure 12 shows the change of vertical earth pressure S2 at pipe top with the increase of surface pressure under different backfill conditions. As the surface pressure increases, the earth pressure above the pipe roof gradually increases. The loose backfill around the pipe and the dense backfill were compared with the use of CLSM backfill. When the surface static pressure is 200 kPa, the earth pressure S2 of the dense backfill and the pipe roof using

CLSM backfill is smaller than that of the loose backfill. The vertical earth pressure S2 of the loose backfilled pipe top was reduced by 5.1% and 66.9%, respectively, compared with the compact backfill and CLSM backfill. The above can be concluded that loose backfill is not conducive to the force of the pipeline. Dense backfill can improve the force characteristics of the pipe, which can reduce the earth pressure at the pipe top. For the backfilling of buried pipelines using CLSM, the pressure of the pipe top can be significantly reduced, and the pipeline can be well protected.

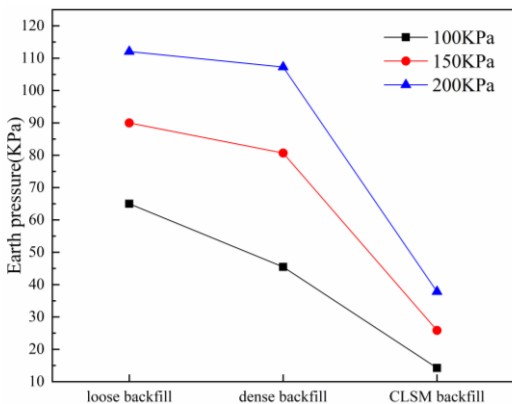

**Figure 12.** Vertical earth pressure at the pipe top.

Figure 13 shows the ratio Sh/Sv (S3/S4, S5/S6) of horizontal earth pressure (S3, S5) to vertical earth pressure (S4, S6) at each monitoring position on the right side of the pipeline Sh/Sv. It can be seen from the figure that for the comparison of dense backfill and CLSM backfill, the ratio of horizontal earth pressure to vertical earth pressure in each monitoring position on the right side of the pipe is constantly decreasing.

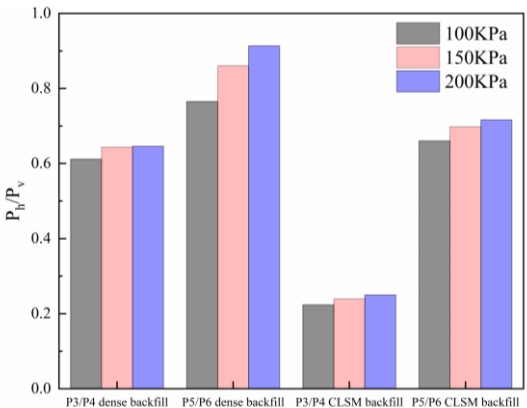

**Figure 13.** Ratio of horizontal earth pressure to vertical earth pressure at each monitoring position on the right side of the pipe.

When the surface pressure is 200 kPa, the S3/S4 of dense backfill and CLSM backfill are 0.64 and 0.25, respectively. Compared with dense backfill, the ratio of horizontal earth pressure to vertical earth pressure for backfilling or using CLSM is reduced by 60.9%. For the earth pressure on the pipe side, CLSM backfilling will reduce the horizontal earth pressure of the pipe and weaken the "earth arch effect" phenomenon of the pipe. The stress of the pipe changes from the pipe–soil interaction to the majority of the solidified CLSM. The stress of the pipeline decreases, which is beneficial to the safety of the pipeline.

The S5/S6 at 15 cm on the right side of the pipe is constantly increasing compared to S3/S4. For compact backfill and CLSM backfill, when the Static pressure is 200 kPa, S5/S6 increases by 42.2% and 184% respectively compared with S3/S4. This is because the vertical pressure on the earth is greater on the side of the pipe than at 15 cm on the right side of the pipe (which is located on the edge of the pressure application).

### 4.4. Radial Displacement of Pipe

Figure 14 shows the radial displacement of the pipe under three different backfill conditions when the surface pressure is 200 kPa. Under three different backfill conditions, the maximum radial displacement of the pipe appears at the top of the pipe. The radial displacement of the pipe top is reduced under dense backfill, and the pipe deformation of the buried pipe is "heart"-like deformation. When the buried pipe is backfilled with CLSM, the deformation of the pipe is extremely small, and the radial displacement of the pipe roof is only 0.6 mm. Compared with the dense backfill, the radial displacement of the pipe top is reduced by 82.9%, which greatly reduces the deformation of the pipe.

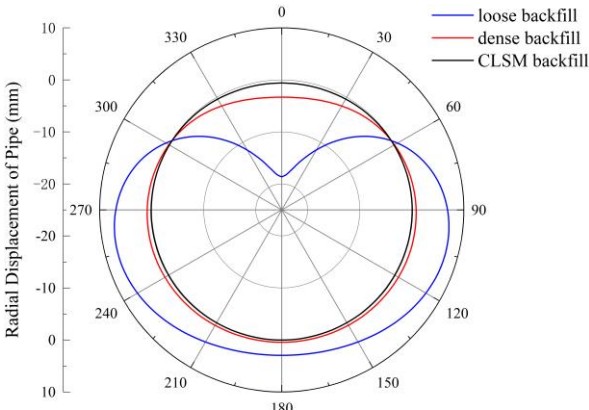

**Figure 14.** Radial Displacement of Pipe.

In the case of loose backfill and dense backfill, the radial displacement of the top of the pipe is 4.7 mm and 3.3 mm, respectively.

### 4.5. Circumferential Strain of Pipe under Different Backfill Conditions

For the three backfill conditions, the annular strain of the pipeline under different surface pressures was recorded. Figure 15 shows the circumferential strain around the pipe during loose backfill. It can be seen from the figure that the circumferential strain of buried pipeline is mainly compressive strain, and the maximum strain is located at the top of the pipe. The surface pressure increases from 100 kPa and 150 kPa to 200 kPa. The circumferential strain around the pipe increases continuously. The circumferential strain of pipe top increased from 0.38% and 0.64% to 0.97%. The strains on the left and right sides of the pipe are basically symmetrical, and tensile strains appear in the range of 45–115° and 245–315° of the pipe. The compressive strain is below the pipe side and the strain is small.

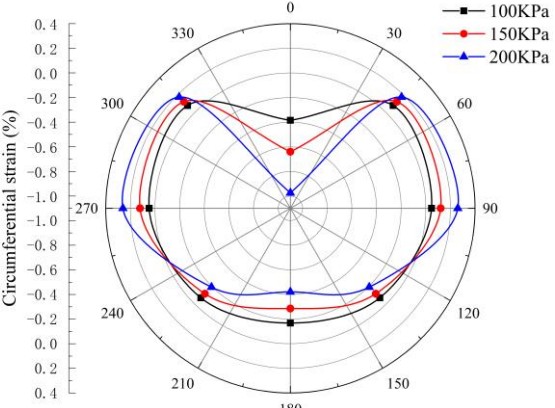

**Figure 15.** Circumferential strain in loose backfill.

Figure 16 show the circumferential strain around the tube during dense backfill and CLSM backfill. The circumferential strain around the pipe increases with the increase of

surface pressure. The surface pressure increased from 100 kPa to 150 kPa and 200 kPa, and the circumferential strain of the pipe top increased from 0.11% to 0.21% and 0.29%. The tensile strain appears in the range of 45–65° and 280–295° of the pipeline, and the range of the pipeline is compressive strain. The range of pipeline compressive strain is smaller than that of loose backfill around the pipe.

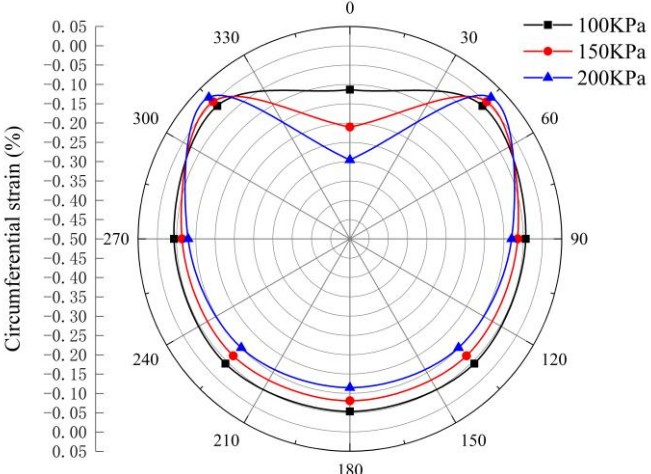

**Figure 16.** Circumferential strain in dense backfill.

Figure 17 shows circumferential strain around the pipe when using CLSM backfill. The surface pressure increased from 100 kPa to 150 kPa and 200 kPa, and the circumferential strain of pipe top increased from 0.02% to 0.04% and 0.059%. Compared with loose backfill and dense backfill, the circumferential strain around the pipe is greatly reduced. When the surface pressure is 150 kPa, the circumferential strain at the pipe top is reduced by 84% and 81%, respectively, compared with loose backfill and dense backfill. The pipe within 360° is compressive strain without obvious maximum and minimum values, and the pipe is uniformly compressed.

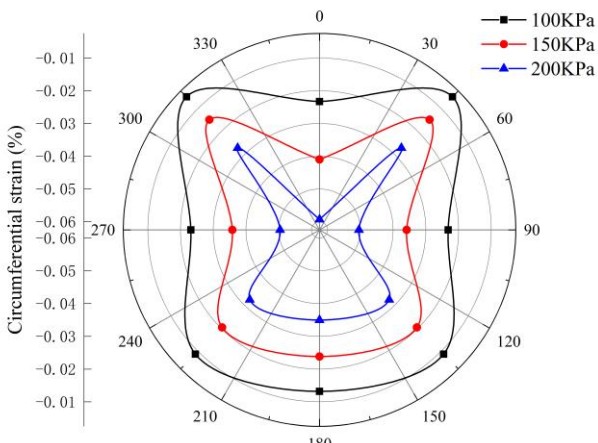

**Figure 17.** Circumferential strain in CLSM backfill.

Figure 18 is the circumferential strain of a pipe under three backfill conditions, when the surface pressure is 200 kPa. The circumferential strain around the buried pipe is greatly reduced by using CLSM backfill. When the surface pressure is 200 kPa, the circumferential strain of the pipe top is 0.97%, 0.29% and 0.059% in the case of loose backfill, dense backfill and CLSM backfill, respectively. Compared with loose backfill and dense backfill, the circumferential strain of CLSM backfill is reduced by 98.9% and 79.7%, respectively. For loose backfill and dense backfill, there is tensile strain around the pipe. Using CLSM for

pipe backfill, the pipeline is no tensile strain, and the compressive strain distribution is relatively uniform. The deformation of the pipe is very small.

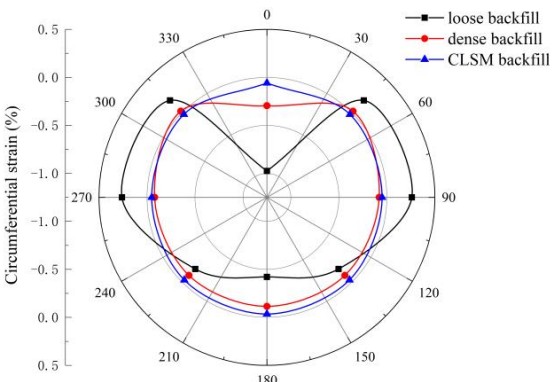

**Figure 18.** Circumferential strain under three backfill conditions.

## 5. Conclusions

1.  In this study, the numerical simulation of the pipe is carried out under three conditions of loose backfill, dense backfill and CLSM backfill. It is concluded that the vertical earth pressure at 15 cm above the buried pipeline increases by 6% compared with the dense backfill. The vertical earth pressure on the pipe top and the horizontal earth pressure on the pipe side are significantly reduced. The horizontal earth pressure on the pipe side is reduced by 39.6% compared with the dense backfill. The CLSM is used for pipeline backfill, which significantly reduces the soil pressure on the pipe top and pipe side and protects the pipeline well.
2.  Using CLSM to backfill the buried pipeline, the bearing force of the pipeline changes from the pipe–soil interaction to the majority of the solidified CLSM. The 'soil arching effect' of pipes almost disappeared. The stress of the pipe is greatly reduced, which is conducive to the operation of the pipeline.
3.  Under three different backfill conditions, the maximum radial displacement of the pipeline appears at the top of the pipe. In the case of dense backfill, the radial displacement of the pipe top is reduced, and the deformation of the buried pipe is 'heart' shaped. When the buried pipe is backfilled by CLSM, the deformation of the pipe is minimal. When the surface pressure is 200 kPa, the radial displacement of the pipe top is only 0.6 mm. Compared with the dense backfill, the radial displacement of the pipe top is reduced by 82.9%, which greatly reduces the deformation of the pipe.
4.  Using CLSM to backfill buried pipeline, the circumferential strain around the pipe is greatly reduced. When the surface pressure is 200 kPa, the circumferential strain of the pipe top decreased by 84.5% and 79.7%, respectively, compared with the loose backfill and dense backfill. The strain around the pipe is very small by using CLSM backfill, and the pressure is uniformly distributed around the pipe.

**Author Contributions:** Conceptualization, H.L. and P.Z.; Data curation, H.L. and Y.X.; Formal analysis, H.L. and Y.X.; Funding acquisition, P.Z.; Investigation, H.L.; Methodology, H.L. and Y.X.; Validation, H.L. and K.L.; Visualization, P.Z.; Writing—original draft, H.L., Y.Z. and P.Z.; Writing—review & editing, H.L. and K.L. All authors have read and agreed to the published version of the manuscript.

**Funding:** This research was funded by Hubei Provincial Natural Science Foundation of China, grant number 2020CFB477.

**Institutional Review Board Statement:** Not applicable.

**Informed Consent Statement:** Not applicable.

**Data Availability Statement:** Not applicable.

**Conflicts of Interest:** The authors declare no conflict of interest.

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
