# Peer review of "Numerical Simulation on Backfilling of Buried Pipes Using Controlled Low Strength Materials"

_applsci, doi:10.3390/app12146901_

Round 1

Reviewer 1 Report

Dear authors,

I found the manuscript interesting and with sound results. I will be very pleased to see it published in its final form in the journal Applied Sciences.

My comments are mainly concern with the way work is presented and less with the contents. Nevertheless, I would like to see them considered in a final version of the manuscript.

#1 Figures and tables captions

The legends (captions) of figures and tables should be revised. I suggest a detailed revision of all the captions, these must be the most complete as possible, because figures should be legible without the need to see the main text. For instance, the tables and figures captions do not refer the study context.

#2 Font size in figures

It would be important to enhance the figures font size, mostly in Figs. 8 and 9 legends.

Regards

Author Response

Dear Editor:

Thank you for your comments. We really appreciate your efforts in reviewing our manuscript.Those comments are valuable and very helpful for this article.We have read through comments carefully and have made corrections. Based on the instructions provided in your e-mail, we uploaded the file of the revised manuscript. Revisions in the text are shown using red highlight and the Word’s revision modefor additions.

#1 Figures and tables captions

The legends (captions) of figures and tables should be revised. I suggest a detailed revision of all the captions, these must be the most complete as possible, because figures should be legible without the need to see the main text. For instance, the tables and figures captions do not refer the study context.

Response 1: Thanks for your comment. We have made changes to the figures and tables captions.

#2 Font size in figures

It would be important to enhance the figures font size, mostly in Figs. 8 and 9 legends.

Response 2: Thanks for your comment. We have enhance the figures font size.

Sincerely,

Reviewer 2 Report

Reviewer Recommendation and Comments for manuscript applsci-1807066-peer-review-v1.pdf with the title: “Numerical Simulation on Backfilling of Buried Pipes Using 2 Controlled Low Strength Materials”, authors: Hao liu, Yiqi Xiao, Kaixin Liu, Youzeng Zhu and Peng Zhang.

This manuscript presents the results of the Numerical Simulation on Backfilling of Buried Pipes Using 2 Controlled Low Strength Materials. From a practical point of view, this is very interesting and valuable research. 

The paper is well organized and systematized in a way that clearly and concisely poses the problem, and defines the subject and methods of research. The introduction consists a lot of data and properly analyzed previous research in the fields of research. The methods are very well described, the results are explained in detail, and the conclusions are adequately derived.

The text is clearly written and the graphical interpretation of the results is clearly presented and provides insight into the results obtained.

The main comments that I find useful for improving the quality of the article are minor and presented below:

Finding 1: Line 9. It is missing a dot at the end of the sentence.

Finding 2: Line 76. It is a missing review of the manuscript true the paragraphs.

Finding 3: Line 105. It is necessary to correct D0 instead of D0 as in formula 3.

Finding 4: Line 106. It is necessary to correct t0 instead of t0 as in formula 3.

Finding 5: Line 107. Figure 2 is not mentioned in the text. Figure 2 does not show any diagrams? Could you check it?

Finding 6: Line 141. Figure 3 is not mentioned in the text. Could you put it in lines 127-132?

Finding 7: Line 151. Could you correct 1.5*1.2 m (width× height) and put 1.5x1.2.

Finding 8: Line 248. Could you check the meaning of “o”?????

The manuscript should be published after minor corrections.

Author Response

Dear Editor:

Thank you for your comments. We really appreciate your efforts in reviewing our manuscript.Those comments are valuable and very helpful for this article.We have read through comments carefully and have made corrections. Based on the instructions provided in your e-mail, we uploaded the file of the revised manuscript. Revisions in the text are shown using red highlight and the Word’s revision modefor additions.

Finding 1: Line 9. It is missing a dot at the end of the sentence

Response 1: We have added it.

Finding 2: Line 76. It is a missing review of the manuscript true the paragraphs.

Response 2: We have modified this.

Finding 3: Line 105. It is necessary to correct Dinstead of D0 as in formula 3.

Finding 4: Line 106. It is necessary to correct tinstead of t0 as in formula 3.

Response 3 ,4 :We have modified this.

Finding 5: Line 107. Figure 2 is not mentioned in the text. Figure 2 does not show any diagrams? Could you check it?

Finding 6: Line 141. Figure 3 is not mentioned in the text. Could you put it in lines 127-132?

Response 5,6: We have added it.

Finding 7: Line 151. Could you correct 1.5*1.2 m (width× height) and put 1.5x1.2.

Response 7 :We have modified this.

Finding 8: Line 248. Could you check the meaning of “o”?????

Response 3 ,4 : I'm sorry. I didn't find it. Can you point it out to me again?

Sincerely,

Reviewer 3 Report

1. Some phrases are not fully understandable. Please correct technical English. 2. Are all the drawings and pictures included in the paper the authors property? 3. Please correct subscripts indexes in the symbols below the presented formulas 4. Were the PE pipes new? This type of material has a "memory" of  applied stresses, which affects the testing of strength parameters. 5. Please provide more details about the testing equipment used for the measurements. 6. Text line no. 127 - CISM what is it? Typo? 7. Text lines no. 127-133 needs to be improved. 8. Figure 3 is not cited in the content 9. How will the strength parameters change with the variable groundwater ordinate? Has such a problem been identified?

Author Response

Dear Editor:

Thank you for your comments. We really appreciate your efforts in reviewing our manuscript.Those comments are valuable and very helpful for this article.We have read through comments carefully and have made corrections. Based on the instructions provided in your e-mail, we uploaded the file of the revised manuscript. Revisions in the text are shown using red highlight and the Word’s revision modefor additions.

1.Some phrases are not fully understandable. Please correct technical English.

Response 1: Thank you for your comments.We have modified in the arctle.

2.Are all the drawings and pictures included in the paper the authors property?

Response 2: The drawings and pictures in the paper are ours property.

3.Please correct subscripts indexes in the symbols below the presented formulas

Response 3: We have modified it.

4.Were the PE pipes new? This type of material has a "memory" of applied stresses, which affects the testing of strength parameters.

Response 4: Thank you for your comments. The PE pipes are new.

5.Please provide more details about the testing equipment used for the measurements.

Response 5: We have added the details about the testing equipment.

6.Text line no. 127 - CISM what is it? Typo?

Response 6: We have modified it to CLSM.

7.Text lines no. 127-133 needs to be improved.

Response 7: We have modified it in the arctle.

8.Figure 3 is not cited in the content

Response 8: We have added it.

9.How will the strength parameters change with the variable groundwater ordinate? Has such a problem been identified?

Response 9: Thank you for your comments. The variable groundwater has an impact on the force on the pipeline. We can think about this interesting direction in the next study.

Sincerely,
